# Radiocarbon Flux Measurements Reveal Mechanistic Insight into Heat-Stress Induction of Nicotine Biosynthesis in *Nicotiana attenuata*

**DOI:** 10.3390/ijms242115509

**Published:** 2023-10-24

**Authors:** Spenser Waller, Avery Powell, Randi Noel, Michael J. Schueller, Richard A. Ferrieri

**Affiliations:** 1Missouri Research Reactor Center, University of Missouri, Columbia, MO 65211, USA; spenserwaller@gmail.com (S.W.); rlnhmh@missouri.edu (R.N.); schuellerm@missouri.edu (M.J.S.); 2School of Natural Resources, University of Missouri, Columbia, MO 65211, USA; 3Division of Plant Science & Technology, University of Missouri, Columbia, MO 65211, USA; 4Chemistry Department, University of Missouri, Columbia, MO 65211, USA; 5Interdisciplinary Plant Group, University of Missouri, Columbia, MO 65211, USA

**Keywords:** heat stress, nicotine biosynthesis, radiocarbon flux analysis, carbon-11, nitrogen-13, protein degradation, L-Asparaginase activity

## Abstract

The effect of high-temperature (HT) stress on nicotine biosynthesis in *Nicotiana attenuata* was examined. Nicotine content was measured in mature leaves, young sink leaves, and in roots from well-watered plants grown at 25 °C as controls and from plants exposed to 38 °C and 43 °C temperatures applied for 24, 48, 72, and 96 h duration. At 38 °C, all leaf nicotine levels were significantly less than control plants for up to 72 h exposure but rose sharply thereafter to levels significantly greater than controls with 96 h exposure. In contrast, plants exposed to 43 °C never exhibited a reduction in leaf nicotine content and showed an increase in content with just 48 h exposure. Using radioactive ^11^CO_2_ and ^13^NO_3_^−^, we found that HT stress reduced both CO_2_ fixation and nitrate uptake. Furthermore, radiocarbon flux analysis revealed that ‘new’ carbon partitioning (as ^11^C) into the ^11^C-radiolabeled amino acid (AA) pool was significantly reduced with HT stress as were yields of [^11^C]-aspartic acid, an important AA in nicotine biosynthesis, and its beta-amido counterpart [^11^C]-asparagine. In contrast, [^12^C]-aspartic acid levels appeared unaffected at 38 °C but were elevated at 43 °C relative to controls. [^12^C]-Asparagine levels were noted to be elevated at both stress temperatures. Since HT reductions in carbon input and nitrogen uptake were noted to impede de novo AA biosynthesis, protein degradation at HT was examined as a source of AAs. Here, leaf total soluble protein (TSP) content was reduced 39% with long exposures to both stress temperatures. However, Ribulose-1,5-bisphosphate carboxylase/oxygenase (Rubisco) which was 41% TSP appeared unaffected. Altogether, these results support the theory that plant proteins other than Rubisco degrade at elevated temperatures freeing up essential AAs in support of nicotine biosynthesis.

## 1. Introduction

Global warming is a real phenomenon that has been documented over the last 100 years [1,2]. With this trend in global warming, mean ambient temperatures are expected to rise 1.0 to 1.7 °C by 2050 and 4.0 to 5.0 °C by the end of the 21st century [3,4,5] at which point, high temperature stress will become one of the most prominent abiotic stresses confronted by the plant kingdom with major impacts to agricultural productivity [6,7,8,9,10].

While most studies concerning climate change have been concerned with plant growth and development, we recognize that many higher plants have evolved various strategies that allow them to adapt to certain environmental stresses. One important adaptation mechanism lies in their ability to synthesize specialized secondary metabolites. Alkaloids represent a group of very specialized secondary metabolites that play a major role in how certain plants can cope with defending themselves against pathogens and generalist herbivores [11,12,13,14]. These substrates often possess unique biological activities and even strong cytotoxic characteristics. *Nicotiana*, a member of the Solanaceous family, produces several pyrimidine-based alkaloids where nicotine is the most dominant in this class of specialized metabolite. Nicotine is biosynthesized primarily in the roots of the plant [15,16,17,18], after which it is transported through the xylem where it accumulates in leaf vacuoles to protect areas of the plant most vulnerable to attack [19,20].

Nicotine biosynthesis has been extensively investigated over the years, leading to a clear picture of its contributing pathways [21]. However, nicotine production is complicated involving multiple steps with two contributing pathways (Figure 1).

Here, the portion involving the formation of a pyrrolidine ring initiates with the formation of putrescine. Putrescine can be produced via two routes: either directly from ornithine involving the ornithine decarboxylase enzyme or indirectly from arginine involving arginine decarboxylase. In the latter, arginine is decarboxylated to agmatine, which is subsequently hydrolyzed to *N*-carbamoylputrescine by agmatine iminohydrolase and further to putrescine by *N*-carbamoylputrescine amidohydrolase. Then, putrescine is converted to *N*-methylputrescine by putrescine *N*-methyltransferase where it is later deaminated oxidatively by *N*-methylputrescine oxidase to 4-methylaminobutanal and spontaneously cyclizes to *N*-methylpyrrolinium cation that contains the pyrrolidine ring.

In the other portion of the pathway aspartic acid is converted to α-iminosuccinic acid, then reacted with glycerol-3-phosphate to form the pyridine ring of quinolinic acid. Nicotinic acid is then formed via nicotinic acid mononucleotide (NaMN) by the same enzymes involved in nicotinamide adenine dinucleotide (NAD) biosynthesis, including aspartate oxidase, quinolinic acid synthase, and quinolinic acid phosphoribosyl transferase. While the exact nature of the metabolite derived from nicotinic acid is not known it is thought that nicotinic acid needs to be reduced to 3,6-dihydronicotinic acid before condensation with *N*-methylpyrrolinium cation to form nicotine.

Many abiotic conditions including water availability [22], light intensity and UV radiation [23], ambient temperature [24,25], as well as other environmental factors including nutrient availability are known to influence plant secondary metabolism. These conditions can induce changes at the level of gene expression, signal hormone transduction, and enzyme activity, which will alter the content of specialized metabolites such as alkaloids [26].

Little is known about the specific effects of heat stress (HT) on the metabolic regulation of plant alkaloids largely because published works are too few and inconsistent. For example, the role of temperature on indole alkaloid content was studied in *Catharanthus roseus* hairy root cultures, where it was shown that by lowering the temperature the level of these alkaloids increased significantly [27]. Specifically, the expression of genes involved in alkaloid biosynthesis was highest in *C. roseus* plants at 12 °C. In that same work, nicotine production in *Nicotiana tabacum* was claimed to be significantly reduced at 30 °C when compared to 25 °C. In contrast to these early observations, more recent studies have shown that elevated temperatures promoted nicotine biosynthesis. In one study, investigators found that HT treatment induced nicotine biosynthesis in *Nicotiana tabacum* which was linked to the rapid accumulation of the suspected signaling agent, hydrogen sulfide. These same studies showed that HT triggered the biosynthesis of jasmonic acid, a plant hormone that is also linked to promoting nicotine biosynthesis [28]. In another study, HT induced the expression of the tobacco transcription factor NtMYC2a that upregulated jasmonic acid biosynthesis by stimulating the transcription of genes associated with JA biosynthesis, including lipoxygenase, allene oxide synthase, allene oxide cyclase, and 12-oxophytodienodate reductase [29]. This action activated the biosynthesis of nicotine and related pyridine alkaloids in tobacco by upregulating the expression of putrescine *N*-methyltransferase 1 (NtPMT1), which encoded putrescine *N*-methyltransferase to catalyze nicotine formation. In yet another more recent study, carbon monoxide signaling was linked to the increased expression of NtPMT1, via promotion of NtMYC2a binding to the G-box region of its promoter all leading to increased nicotine production with HT stress [30].

While these past studies provide compelling arguments for HT induction of the putrescine pathway in nicotine biosynthesis, there has not been a study that addresses whether HT stress impacts the aspartate pathway contributing to nicotinic acid biosynthesis and nicotine production. To explore the underlying mechanisms of metabolic regulation, we undertook a series of studies in *Nicotiana attenuata* where plants were subjected to temperature stresses at 38 °C and 43 °C and examined for temporal changes in tissue nicotine content. Furthermore, we leveraged the power of radiocarbon flux measurements using radioactive carbon-11 to gain insight into the effects of HT stress on aspartate regulation.

## 2. Results

### 2.1. Acute and Chronic Heat Stress Alters Tissue Nicotine Content

We examined the effects of heat stress at 38 °C (Figure 2A–C) and 43 °C (Figure 2D–F) applied across timelines spanning 24 h to 96 h of exposure at each temperature. Data were collated showing how temperature and time altered the nicotine content in mature leaves, young sink leaves, and in root tissues.

Mature leaf nicotine content was initially reduced at 38 °C to 1.74 ± 0.12 mg g^−1^DW with 24 h exposure and to 1.92 ± 0.46 mg g^−1^DW with 48 h exposures relative to 25 °C control levels of 3.64 ± 0.54 mg g^−1^DW. However, these levels increased sharply and significantly by 96 h exposure to 6.66 ± 0.58 mg g^−1^DW. Young sink leaves possessed less nicotine than fully expanded mature leaves harvested from the same plants. Even so, young leaves exhibited the same behavior as their mature leaf counterparts when exposed to 38 °C. Here, levels dropped significantly to 0.45 ± 0.08 mg g^−1^DW with 24 h exposure and 0.56 ± 0.15 mg g^−1^DW with 48 h exposures relative to 25 °C control levels of 1.82 ± 0.19 mg g^−1^DW but then started to rise slightly with 72 h exposure to 1.56 ± 0.16 mg g^−1^DW and increased even more sharply and significantly to 4.29 ± 0.30 mg g^−1^DW with 96 h exposure.

In contrast to data collected at 38 °C, mature leaf nicotine content of plants exposed to 43 °C never dropped below control values of 3.64 ± 0.54 mg g^−1^DW but rather rose gradually to 5.49 ± 0.77 mg g^−1^DW with 24 h exposure, to 5.84 ± 1.31 mg g^−1^DW with 48 h exposure, to 7.38 ± 0.96 mg g^−1^DW with 72 h exposure, and finally to 11.34 ± 0.71 mg g^−1^DW with 96 h exposure. We also note that the final mature leaf nicotine content with 96 h exposure to 43 °C temperature was significantly higher than that at 38 °C.

Identical trends were observed at 43 °C when examining changes in nicotine content in the young sink leaves. Here, a gradual but significant rise in content was observed as a function of exposure time to a final level of 6.42 ± 0.44 mg g^−1^DW at 96 h exposure. Unlike their mature leaf counterparts, this rise in nicotine content at 43 °C was not significantly different to that at 38 °C.

Roots exhibited identical behavior at both elevated temperatures where significant downward trends in nicotine content were observed with increasing exposure from 1.34 ± 0.19 g^−1^DW to 0.74 ± 0.08 g^−1^DW.

### 2.2. Effect of Heat Stress on ^11^CO_2_ Fixation and ^13^NO_3_^−^ Uptake

Using radioactive ^11^C administered to plants as ^11^CO_2_ we examined the effect of applied heat stresses on carbon fixation. Radiotracer was administered to plants after 48-h exposure to heat stress temperatures of 38 °C and 43 °C and compared to levels for control plants grown at 25 °C. Results in Figure 3A show that leaf fixation of radiotracer decreased using these treatments. Specifically, fixation decreased significantly from 64.97 ± 2.66 percent of ^11^C radioactivity within the pulse (% fixation) for 25 °C control plants to 50.28 ± 3.70% at 38 °C and 45.46 ± 2.90% at 43 °C. In the ^13^NO_3_^−^ uptake studies, data were normalized to total plant mass and presented as the amount of ^13^N radioactivity found in plant tissue. Here, results showed that nitrate uptake decreased significantly with HT stress from 7.8 × 10^6^ ± 8.0 × 10^5^ cpm for control plants to 5.8 × 10^6^ ± 8.2 × 10^5^ cpm at 38 °C and 4.0 × 10^6^ ± 8.4 × 10^5^ cpm at 43 °C.

### 2.3. Effect of Heat Stress on ‘New’ Carbon Partitioning (as ^11^C) into Amino Acids

The same tracer provided insight into the state of de novo amino acid biosynthesis. Results in Figure 4A showed that ‘new’ carbon partitioning into the total free amino pool decreased significantly to levels of 3.14 ± 0.56% and 2.78 ± 1.06% of total ^11^C activity assimilated by the leaf at 38 °C and 43 °C, respectively, relative to control levels of 6.10 ± 0.99%. Likewise, results in Figure 4B showed that ‘new’ carbon partitioning into [^11^C]-aspartic acid decreased significantly to levels of 0.69 ± 0.07% and 0.41 ± 0.28% of total ^11^C radiotracer assimilated by the leaf at 38 °C and 43 °C, respectively, relative to control levels of 1.49 ± 0.12%. Similarly, levels of [^11^C]-asparagine, the amido counterpart to aspartic acid, dropped (Figure 4C) to levels of 0.23 ± 0.03% and 0.31 ± 0.09%, respectively, relative to control levels of 1.17 ± 0.39%. Here, it was important to monitor fluxes of both amino acids since the action of asparaginase enzyme can convert asparagine to aspartic acid.

### 2.4. Effect of Heat Stress on the Endogenous Concentrations of [^12^C]-Aspartic Acid and [^12^C]-Asparagine

Here, we examined the effect of heat stress on the endogenous pools of aspartic acid and its counterpart asparagine in the leaf tissue and root tissues. Data were collected with 48 h exposure to heat stress temperatures of 38 °C and 43 °C and compared to levels for control plants grown at 25 °C. Results in Figure 5A showed that the endogenous leaf concentration of aspartic acid at 38 °C was 3.46 ± 1.79 µmol per gram fresh weight of tissue (µmol g^−1^FW) which was not significantly different from the 5.63 ± 0.49 µmol g^−1^FW level observed for controls. In contrast, the leaf level of aspartic acid rose significantly to at 43 °C to 28.52 ± 1.60 µmol g^−1^FW. Results in Figure 5B showed that the endogenous leaf concentration of asparagine rose significantly with applied heat stresses to 40.65 ± 4.39 µmol g^−1^FW and 102.85 ± 13.35 µmol g^−1^FW for 38 °C and 43 °C, respectively, relative to a 24.46 ± 5.14 µmol g^−1^FW level for controls.

In sharp contrast to leaves, root tissues presented a very different picture of how heat stress affected aspartic acid and asparagine levels. Results in Figure 5C showed that endogenous concentrations of aspartic acid dropped significantly to levels of 1.43 ± 0.28 µmol g^−1^FW and 1.01 ± 0.38 µmol g^−1^FW at 38 °C and 43 °C, respectively, relative to control levels of 4.91 ± 0.54 µmol g^−1^FW. In Figure 5D that the endogenous concentration of 38.22 ± 2.58 µmol g^−1^FW for asparagine at 38 °C was significantly higher than the 30.37 ± 4.97 µmol g^−1^FW level of controls but then turned down sharply and significantly to 2.31 ± 0.55 µmol g^−1^FW at 43 °C. Overall, concentrations of aspartic acid and asparagine in roots were significantly less than those see in mature leaves where these amino acids are synthesized.

### 2.5. Effect of Heat Stress on Leaf Total Protein and Rubisco Protein Content

Finally, we examined the influence of heat stress on leaf soluble total protein content and leaf Rubisco content. Results in Figure 6A,B showed significant downward trends in the total soluble protein (TSP) content dropping 39% from a level of 116.68 ± 2.95 mg protein per gram dry weight (mg g^−1^DW) for 25 °C controls to 72.07 ± 3.380 mg g^−1^DW and 70.77 ± 3.71 mg g^−1^DW with 96 h exposures to applied stresses at 38 °C and 43 °C, respectively. Contrary to this, leaf Rubisco which amounted to ~41% TSP decreased only slightly from the 47.22 ± 2.06 mg g^−1^DW level for 25 °C controls, but that downward trend in content with applied heat stress and exposure was not significant.

## 3. Discussion

The present work brings to light answers to why there has been so much inconsistency in past reporting of HT stress effects on nicotine biosynthesis [27,28,29,30]. Our work points to both temperature of the HT stress and the duration of exposure as equally important factors to consider when reporting results.

In agreement with past HT stress studies [31,32,33,34], our work showed that leaf CO_2_ assimilation decreased significantly with applied stresses. Typically, assimilation of nitrogen in higher plants will be balanced by the plant’s carbon input or photosynthetic capacity [35]. This behavior was clearly demonstrated in the present work where HT stress reduced ^13^NO_3_^−^ uptake in plants. As a corollary to this, our past work has also demonstrated that nitrogen limitation during growth will downregulate leaf photosynthesis [36]. Thus, with the decreased availability of ‘new’ carbon and nitrogen resources, it is not so surprising that our radiocarbon flux measurements showed a significant reduction in radiolabeled amino acids.

Shown pictorially in Figure 7, the reduction in CO_2_ fixation seen in the present work with applied HT stress caused a rebalancing of plant resources and a reduction in NO_3_^−^ uptake. Taken together, this rebalancing of resources caused the reduction in the de novo biosynthesis of essential amino acids that serve nicotine biosynthesis.

In most higher plants, asparagine, and glutamine are the most abundant amino acids found in the plant vasculature although all amino acids are capable of transporting from sites of their biosynthesis to distal tissues where they are used in plant metabolism [37,38,39,40,41,42]. With shortfalls in the supply of [^11^C]-amino acids we questioned what was supplying the aspartic acid needed to support increased nicotine biosynthesis from HT stress.

Our examination of the endogenous pools of leaf aspartic acid and asparagine revealed a different story from the radiotracer results with possible insight into a mechanism for nitrogen remobilization. Here, 38 °C HT stress did not influence the endogenous pool of leaf aspartic acid, although the 43 °C stress did elevate its content significantly relative to controls. Furthermore, the pool of leaf asparagine was significantly elevated with the application of both HT stresses. Since de novo biosynthesis of these amino acids were downregulated with HT stress, there needs to be another source maintaining and even increasing their content in tissues.

It is well established that many plant proteins will undergo degradation when subjected to HT stress [10,43,44,45]. In turn, this action will free up many α-amino acids that comprise the protein backbone. We examined the extent of protein degradation in the present work by measuring the content of both leaf total soluble protein (TSP) as well as Rubisco protein as a function of applied HT stresses and the different exposure times to these stresses. As noted earlier, TSPs decreased by nearly the same amount when subjected to either 38 °C or 43 °C stresses and these reductions became significant with greater than 24 h exposures. Interesting, Rubisco, which is typically the most abundant leaf protein (comprising 41% TSP in the present work), was not seen to degrade significantly with temperature nor exposure time. However, this behavior is consistent with prior reports of HT stress on Rubisco protein content [33,34] where reduction in photosynthetic capacity was attributed to loss of Rubisco activase and not to Rubisco protein content. Thus, the supply of essential ‘free’ α-amino acids supporting nicotine biosynthesis was rapid and identical for both stress temperatures and derived from ‘other’ regulatory proteins, and not Rubisco.

Even so, the rapid turnover of protein to free up α-amino acids at both HT stresses does not explain the initial drop observed in foliar nicotine content with short exposures at the lower 38 °C temperature. Hence, there must be another regulatory control mechanism underpinning the process. Some insight into its nature comes from our examination of root endogenous pools of aspartic acid and asparagine. As expected, the increased tug on the root aspartic acid pool when the nicotine biosynthesis was upregulated by HT stress manifested in a significant reduction in its root content. We suspect that the supply train feeding this aspartic acid pool may derive, in part, from the competitive conversion of recently ‘freed’ asparagine from protein degradation to aspartic acid. This action is driven by L-Asparaginase (*EC 3.5.1.1*), an enzyme that catalyzes asparagine’s hydrolysis yielding aspartic acid and ammonia in the process [46]. On closer inspection, our results show that root asparagine at 38 °C was significantly higher than control levels. As leaf protein degradation was prominent even at this temperature, foliar asparagine, ‘freed’ from this process, likely translocated to the roots causing the elevated levels observed in those tissues. We noted earlier that asparagine is one of the most abundant mobile amino acids found in plant vasculature; so, it is likely that what was remobilized in leaf tissue would supply the roots with this amino acid. Interestingly, at 43 °C, asparagine levels suddenly dropped to levels significantly less than controls. To rationalize this drastic shift in metabolism, we suggest that L-Asparaginase (ASNase) function may be strongly coupled to temperature and its state of activation may have accounted for the delayed response in root asparagine conversion to aspartic acid, and ultimately accounted for the delayed response observed in the HT induction of nicotine biosynthesis at 38 °C. Indeed, because of its high commercial value [47] with applications in the food industry and as an anti-cancer drug in the treatment of lymphoblastic leukemia, researchers have examined the thermostability of ASNase and found its activity is strongly correlated with temperature [48].

In future studies, we plan to test the hypothesis that the lag in ASNase function with time and temperature was the reason underpinning the initial decline in foliar nicotine content. From past work [36], the application of transgenic methods to transfer specific nucleic acid sequences from potato- and soybean-encoding ASNase were shown to generate viable ASNase overexpressing lines in *Nicotiana* (35S: PotASNase 1–4; 35S: SoyASNase 2–38) that exhibited significant reductions in asparagine content. By subjecting these lines to HT stress, we expect the initial decline in foliar nicotine at 38 °C to disappear, matching instead the trend observed at higher temperature.

Lastly, we note that while nicotine production appeared to ramp up over time with exposure to applied HT stresses, root nicotine content continually dropped. Past work has identified several transport proteins in *N. tabacum* that likely play an important role in the cellular trafficking of nicotine [19,20]. Of course, once in the xylem, this alkaloid will quickly travel to aboveground tissues relying on the water transpiration stream to drive its translocation aboveground. Our data suggests that roots do not retain much of the nicotine alkaloid they produce but rather quickly ship it aboveground. To test this theory, we are presently working on radiolabeling nicotine with ^11^C and plan to utilize the radiotracer to examine the influence of temperature on nicotine transporter function.

## 4. Materials and Methods

### 4.1. Plant Growth

Seeds of *Nicotiana attenuata* Torr. ex S. Watson (Accension W6 59297; USDA Agricultural Research Services, Pullman, WA, USA) were germinated in ProMix (Strawberry Hill Farms, Columbia, MO, USA) and after 10 days seedlings were transplanted to individual 3-inch pots filled with ProMix and supplemented with Osmocote™ Smart Release Plant Food (Scotts Co., Marysville, OH, USA) for continued growth in a commercial growth chamber (Model PGC-15, Percival Scientific, Perry, IA, USA) under 12 h photoperiods at 500 μmol m^−2^ s^−1^ light intensity, constant temperature of 25 °C, and relative humidity (RH, 60%). Control and heat stress studies were conducted in 5-week-old plants. When applying heat stress, plants were transferred from their control growth chamber to a second chamber that was operated either at 38 °C or 43 °C while maintaining the same light intensity and relative humidity as the control chamber. Stress treatment duration was for 24, 48, 72, or 96 h. Pots were placed in trays that held a surplus of water that was wicked up through the ProMix to maintain a constant well-watered status throughout the stress treatment. During treatment, plant leaves did not exhibit visible signs of damage, or necrosis.

### 4.2. Nicotine Analysis

After stress treatment, mature leaves and young sink leaves were harvested from individual plants and air dried for 1 week at 70 °C. Roots were also harvested, washed rigorously in water, and dried at 70 °C. Dried tissues were placed in pre-weighed 1.5 mL Eppendorf™ tubes equipped with a stainless-steel ball bearing. Sufficient tissue was introduced to each sample tube to ensure at least 500 mg of ground tissue would be available after the grinding process. Samples were ground to a powder using a ball mill grinder (Retsch, Inc., Newtown, PA, USA) and exact tissue masses were recorded. Ground tissue was mixed with 0.5 mL 1 M NaOH and 0.5 mL petroleum ether and the contents were shaken for 15 min using the ball mill grinder adjusted to a lower oscillation setting (10 oscillation s^−1^). The contents were centrifuged at 14,000× *g* for 5 min and the upper organic layer was removed. The organic extraction process was repeated twice more to yield a total of 1.5 mL of petroleum ether when all the extractions were combined. The contents were placed in a vacuum centrifuge (Labconco, Inc., Kansas City, MO, USA) set for 14,000× *g*, 80 °C, and a 30 min duration. This process evaporated the petroleum ether. Samples were then reconstituted in 100 µL methanol, vortex mixed, and 1 µL aliquots were injected into a flame-ionization Hewlett Packard 5890A gas chromatograph equipped with a 30 m long capillary column, 0.53 mm I.D., 0.25 µm thick film of Rtx^®^-Wax (Restek, Inc., Bellefonte, PA, USA) with a 10 mL min^−1^ helium flowrate and a 1:50 sample split. At injection the initial column temperature was maintained at 100 °C and was programmed to increase to 220 °C at a 3 °C min^−1^ rate. The sample injector was maintained at 250 °C and the flame ionization detector at 350 °C. Chromatographic peaks for nicotine were recorded at a 7.4-min retention time and quantified using PeakSimple™ chromatography software v4.88 (SRI, Inc., Torrance, CA, USA) against a commercial nicotine standard (Sigma Aldrich, St. Louis, MO, USA).

### 4.3. Production and Administration of Radioactive ^11^CO_2_ and ^13^NO_3_^−^

^11^CO_2_ (t_½_ 20.4 min) was produced on the GE PETtrace Cyclotron located at the Missouri Research Reactor Center using high-pressure, research-grade N_2_ gas target irradiated with a 16.4 MeV proton beam to generate ^11^C via the ^14^N(p,α)^11^C nuclear transformation [49,50]. The ^11^CO_2_ was trapped on the molecular sieve, desorbed, and quickly released into an air stream at 200 mL min^−1^ as a discrete pulse of 740 MBq activity for labeling a leaf affixed within a 5 × 10 cm lighted (560 µmol m^−2^ s^−1^) leaf cell to ensure a steady level of fixation. The load leaf affixed within the cell was pulse-fed ^11^CO_2_ for 1 min, then chased with normal air for the duration of exposure [51]. A PIN diode radiation detector (Carroll Ramsey Associates, Berkeley, CA USA) attached to the bottom of the leaf cell enabled continuous measurement of radioactivity levels within the cell during the initial pulse and in the minutes directly following to give information on ^11^CO_2_ fixation. The attached leaf was incubated for 20 min after which exposed tissue was harvested for analysis.

Radioactive ^13^N (t_½_ 9.97 m) was produced via the ^16^O(p,n)^13^N nuclear transformation by irradiating natural abundance water with 16.4 MeV protons [49,50]. The ^13^N was recovered as carrier-free ^13^NO_3_^−^ where 1850 MBq of radioactivity was administered to plant roots in their growing pots using 25 mL of deionized water to uniformly disperse the radiotracer. Radiotracer was allowed to incubate for 90 min after which plants were removed from their pots, the roots were washed 3 times in deionized water to remove surface radioactivity, and radioactivity was counted for ^13^NO_3_^−^ assimilated into tissue using a gamma counter. Total tissue was weighed to normalize ^13^NO_3_^−^ uptake data to a fixed mass.

### 4.4. [^11/12^C]-Aspartic Acid and [^11/12^C]-Asparagine Analyses

Exposed leaf tissue (~500 mg) was flash frozen in liquid nitrogen, ground to a fine powder in pre-weighed 1.5 mL Eppendorf™ tubes using a ball mill grinder, then weighed for exact tissue mass and extracted in 1 mL methanol: water (60:40 *v*/*v*) using a cell disruptor (Branson, Bransonic 32; Sigma-Aldrich Corp) set at 100% amplitude for 2 min duration. Afterwards, samples were centrifuged at 14,000× *g* for 2 min to separate insoluble and soluble portions. Supernatant was removed and placed in a separate Eppendorf™ where a 20 µL aliquot was removed for counting ^11^C-radioactivity using a NaI gamma counter. The pellet was also counted for ^11^C-radioactivity on the same instrument. Radioactivity data were decay-corrected back to the end of bombardment, or end of the cyclotron beam. A visual of the workflow can be found in Figure 8. The combined radioactivity of the fraction-corrected soluble extract plus pellet provided a measure of total ^11^C-radioactivity from which to base [^11^C]-amino acid yields.

[^11/12^C]-aspartic acid and [^11/12^C]-asparagine were analyzed using pre-column OPA(*o*-phthaldehyde) derivatization and quantified using gradient radio HPLC (Sonntek, Inc. Upper Saddle River, NJ, USA) via a Phenomenex Luna 5µm C18(2), 100 mm × 4.6 mm inner diameter column heated to 30 °C and mobile phase system comprised of Solvent A (30 mM KH_2_PO_4_; 0.4% (*v*/*v*) tetrahydrofuran; pH 6.8), Solvent B (1:1 acetonitrile: water) starting at 100% of A at injection and ramping to 50:50 of A:B within 20 min at a flow rate of 1.5 mL min^−1^. This method enabled baseline separation of aspartic acid and asparagine in less than 6 min but left many of the remaining amino acids unresolved. On-line fluorescence detection (340 nm excitation, 450 nm emission; Hitachi LaChrom Elite L-2485; Sonntek, Inc.) was used for the quantification of the OPA-derivatized peaks that were associated with [^12^C]-aspartic acid and [^12^C]-asparagine and a NaI gamma radiation detector (Eckert and Ziegler, Inc., Wilmington, MA, USA) enabled direct measurement of the [^11^C]-labelled amino acids. Data were acquired using PeakSimple™ chromatography software v4.88 (SRI, Inc., Torrance, CA, USA) and compared to standard calibration curves constructed for aspartic acid and asparagine providing absolute amounts for these amino acids in units of µmol per gram fresh weight (g^−1^FW) tissue. Radioactive metabolite peaks for [^11^C]-aspartic acid and [^11^C]-asparagine were quantified using the same software, corrected for radioactive decay, and corrected for differences in efficiencies of the gamma counter NaI detector and the HPLC’s NaI detector. As noted earlier, the other radioactive amino acids were unresolved with our HPLC method but were recorded as a sum of their contributions to the radiolabeled amino acid pool. After corrections, individual [^11^C]-aspartic acid and [^11^C]-asparagine data were reported as percent total ^11^C-radioactivity within the plant as was the total ^11^C-labeled amino acid fraction reported as percent total ^11^C-radioactivity.

### 4.5. Protein Analysis

Mature leaf tissues were harvested after the different stress treatments were applied and tissues were stored at ™80 °C until analyzed. For protein extraction, samples were placed in a freeze-dryer (FreezeZone 1 Dryer, Labconco Corp., Kansas City, MO, USA) and lyophilized. Once dried, these samples were ground to a fine powder using the ball mill grinder, weighed in 100 mg amounts (±1 mg), and placed into 1.5 mL Eppendorf™ tubes. Leaf pigments and other confounding leaf metabolites were first removed via extraction with samples using 1 mL cold acetone (5% trichloroacetic acid, *w*/*v*) following published procedures [52]. Samples were then sonicated for 15 min and centrifuged at 14,000× *g* for 10 min to separate the protein pellet. Once the supernatant was removed, the pellet was washed three more times with cold acetone to remove lingering trichloroacetic acid. Samples were dried at ambient temperature under vacuum. Once dried, total proteins were extracted in 1 mL of 50 mM Tris–Cl buffer (pH 8.8) containing (1% *v*/*v*) plant protease inhibitors (Plant Protease Inhibitor Cocktail, Sigma Aldrich). Liquefied tissues were clarified via centrifugation at 14,000× *g* for 10 min, and supernatant was removed and placed into a clean tube for analysis.

Following a published procedure [53] describing selective Rubisco fractionation, we mixed 500 µL of protein extract with an equal volume (50% *v*/*v*) of 100 mM CaCl_2_·2H_2_O and sodium phytate (Sigma Aldrich, St. Louis, MO, USA) to selectively precipitate the Rubisco protein. After addition of the fractionation agents, gentle mixing, and incubation, solutions were centrifuged at 14,000× *g* for 10 min and the supernatant removed. The protein precipitate was washed with cold acetone and resuspended in 50 mM Tris–Cl buffer (pH 8.8) containing (1% *v*/*v*) plant protease inhibitors. The solution was clarified via centrifugation and the resolubilized Rubisco protein collected for analysis.

Following published procedures for the Bradford Assay (Sigma Aldrich Technical Report B6916), 100 µL aliquots of protein extract were removed from each sample tube and mixed with 3 mL of Bradford Reagent (Sigma Aldrich, Inc., St. Louis, MO, USA) in 10 mL glass vials [54]. Samples were incubated at room temperature for 45 min and then transferred to optical cuvettes to measure their absorbance at 595 nm using an Evolution 201 UV/VIS spectrophotometer (ThermoFisher Scientific Inc., Waltham, MA, USA). Absorbances were correlated using a standard curve created from measurements using a commercial protein standard (1 mg mL^−1^) of bovine serum albumin (Sigma Aldrich, Inc., St. Louis, MO, USA).

### 4.6. Statistical Analysis

Data were subjected to one-way analysis of variance (ANOVA) using SigmaPlot 14.5. Tukey’s HSD test was used for post hoc correction of comparisons of stress treatments to unstressed control plants. Significance was set at a level of *p* < 0.05.

## 5. Conclusions

In summary, this work clearly demonstrates that nicotine biosynthesis is inducible with HT stress. However, regulatory control of the nicotine pathway, at least regarding aspartic acid’s contribution, is not straight forward. With short exposure times at 38 °C foliar tissues had less nicotine content than 25 °C controls, but this level increased over time of exposure to the stress. At 43 °C, foliar nicotine levels were consistently elevated over those of control plants. Our application of radiocarbon flux analysis provided mechanistic insight into the reasons for this difference. With HT stress, plant C input declines significantly. Plants adapt to this by rebalancing their nitrogen uptake to their carbon input. This fact was evident by the significant decrease seen in ^13^NO_3_^−^ uptake. Hence, there is less nitrogen available to supply amino acid biosynthesis which accounted for the reduction in radiolabeled amino acids. Our data suggests that, in order to supply the nicotine biosynthesis with essential amino acids, protein degradation at HT likely is the source for remobilized N. We also note that the delayed response in the HT induction of nicotine biosynthesis with short exposure to 38 °C temperatures was likely due to the slower response of ASNase to temperature causing a lag in the conversion of asparagine to aspartic acid in the root tissues.

From an ecological perspective, the stresses imposed by climate change will naturally weaken a plant. To survive, plants might elevate their defense posture. At least for *N. attenuata,* we see an upregulation in alkaloid biosynthesis with increased distribution to vegetative tissues that can help protect the plant from chewing herbivores [11,12,13,14] while in a weakened state of a carbon and nitrogen resource deficiency. However, an ecological paradox becomes evident considering other published works that have shown that pollinator behavior can be altered by low levels of nicotine in floral nectar. Here, researchers have shown that there is an increased frequency of visitation to flowers containing small amounts of nicotine by hummingbirds [55] and bees [56] which could promote plant reproduction by boosting pollination. This begs the question whether nicotine-producing plants do so exclusively to defend themselves against attack, or in this case where HT stress reduces plant fitness, thereby serving as a mechanism to promote the survivability of their progeny.

## Figures and Tables

**Figure 1 ijms-24-15509-f001:**
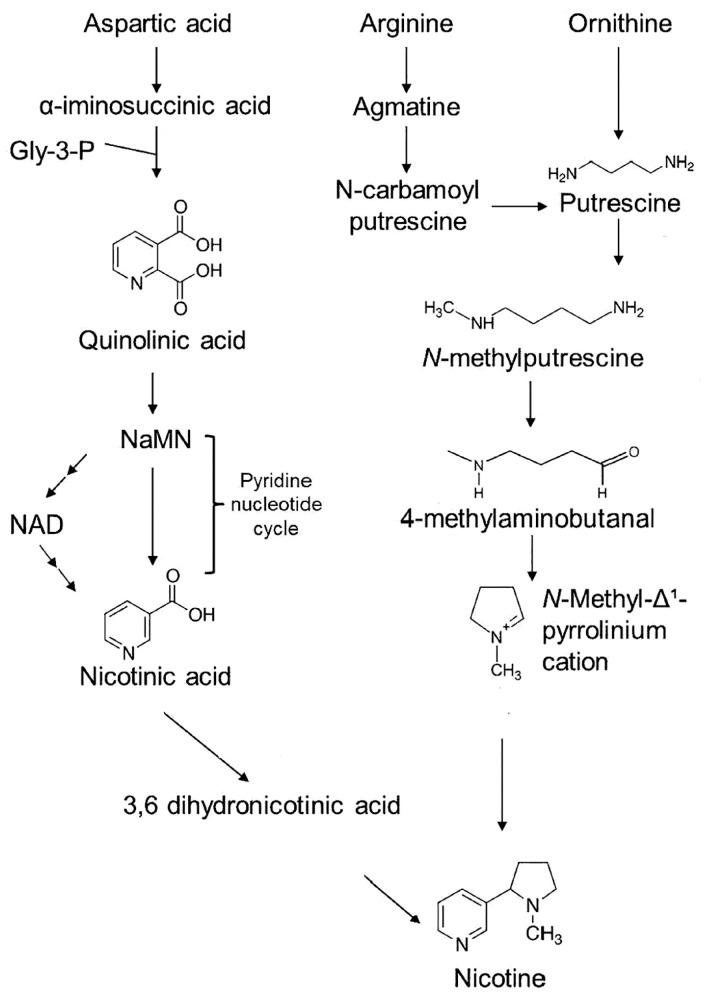
Biosynthetic pathway for nicotine production [21].

**Figure 2 ijms-24-15509-f002:**
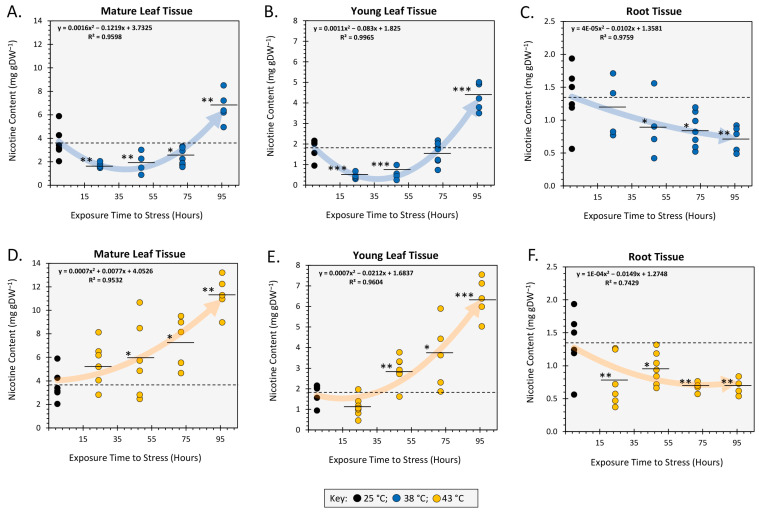
The effect of heat stress on nicotine content presented as milligram of nicotine per gram dry weight tissue (mg g^−1^DW). Heat stresses were examined at 38 °C (Panels **A**–**C**) and 43 °C (Panels **D**–**F**) applied across exposure timelines spanning 24 h to 96 h. Tissues examined included mature expanded leaves, young developing leaves, and roots. All data points are shown for at least N = 6 replicates for each timepoint. The 25 °C control data mean value is shown as the dashed line in each panel. Mean values for each treatment dataset are presented as black horizontal bars with levels of significance set as *, *p* < 0.05; **, *p* < 0.01; and ***, *p* < 0.001. Best fit trendlines were applied to data within each panel where the equations and the goodness of fit are displayed.

**Figure 3 ijms-24-15509-f003:**
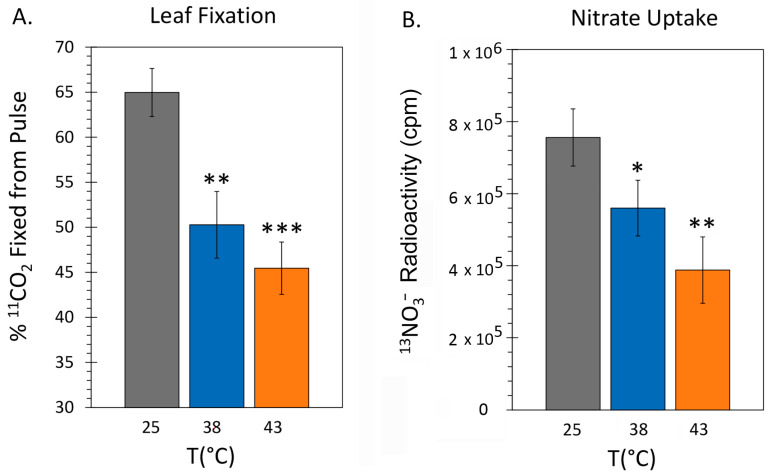
The effect heat stresses of 38 °C and 43 °C applied for 48 h on ^11^CO_2_ fixation in mature leaf tissue and whole-plant ^13^NO_3_^−^ uptake. (Panel **A**): percent ^11^CO_2_ fixation (leaves were normalized to a standard mass). Data bars represent mean values ± SE percent of total ^11^C activity in the pulse for N = 3–5 replicates at each temperature. (Panel **B**): amount of ^13^N radioactivity presented as counts per minute (cpm) assimilated by the plant as ^13^NO_3_^−^ over 90 min of incubation. Data bars represent mean values ± SE for N = 3 replicates at each temperature. Levels of significance were set as *, *p* < 0.05; **, *p* < 0.01; and ***, *p* < 0.001.

**Figure 4 ijms-24-15509-f004:**
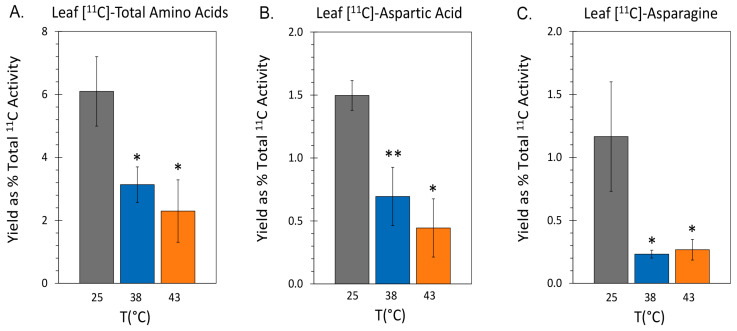
The effect heat stresses of 38 °C and 43 °C applied for 48 h on the metabolic partitioning of ‘new’ carbon (as ^11^C) in mature leaf tissue. All data bars represent mean values ± SE percent of total ^11^C activity assimilated by the leaf through ^11^CO_2_ fixation for N = 3–5 replicates. (Panel **A**): percent of ^11^C partitioning into the total amino acid pool. (Panel **B**): percent of ^11^C partitioning into [^11^C]-aspartic acid. (Panel **C**): percent of ^11^C partitioning into [^11^C]-asparagine. Levels of significance set as *, *p* < 0.05 and **, *p* < 0.01.

**Figure 5 ijms-24-15509-f005:**
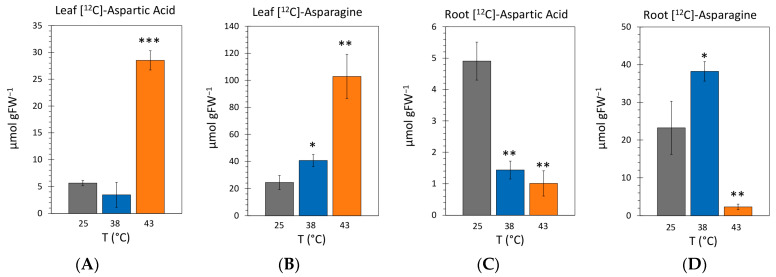
The effect heat stresses of 38 °C and 43 °C applied for 48 h on the endogenous concentrations of aspartic acid and asparagine amino acids in mature leaves and roots. All data bars represent mean values ± SE in micromoles of metabolite per gram fresh weight of tissue (µmol g^−1^FW) for N = 3–5 replicates. (Panels **A**,**B**): reflect levels of aspartic acid and asparagine in leaves. (Panels **C**,**D**): reflect levels of aspartic acid and asparagine in roots. Levels of significance set as *, *p* < 0.05; **, *p* < 0.01; and ***, *p* < 0.001.

**Figure 6 ijms-24-15509-f006:**
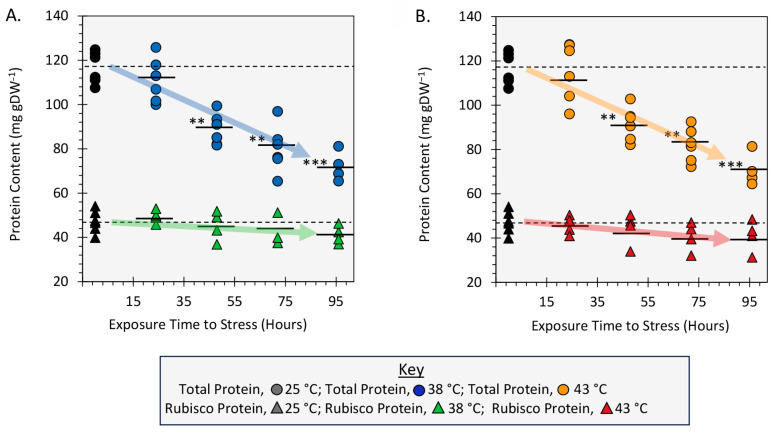
The effects of heat stresses of 38 °C (Panel **A**) and 43 °C (Panel **B**) spanning a timeline of exposure from 24 h to 96 h on total soluble leaf protein content and Rubisco protein content presented in milligrams protein per gram dry weight tissue (mg g^−1^DW). All protein data points are shown for N = 3–6 replicates at each timepoint. Mean values for total protein content and Rubisco protein content for 25 °C control plants are shown as horizontal dashed lines. Mean values from applied heat stresses are also shown by the solid black horizontal bars with levels of significance shown as **, *p* < 0.01 and ***, *p* < 0.001.

**Figure 7 ijms-24-15509-f007:**
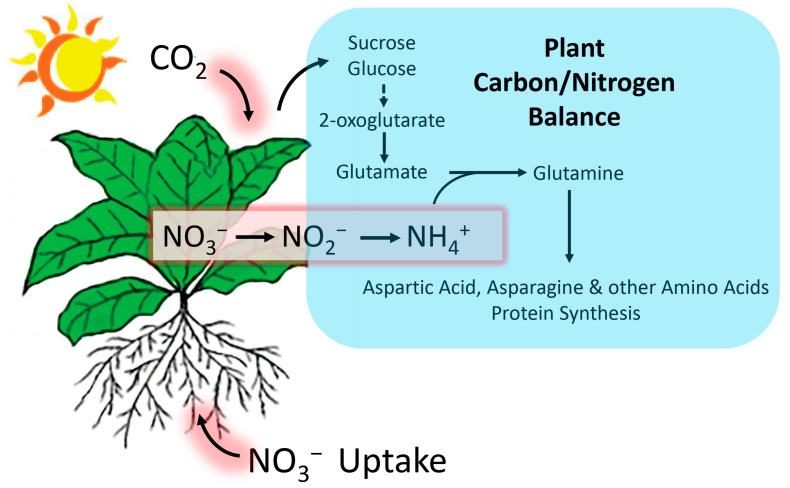
Plant tight coordination of carbon and nitrogen metabolism. CO_2_ fixation and NO_3_^−^ uptake occur in the leaf and the root systems, respectively. 2-oxoglutarate, an important intermediate product of carbon metabolism, serves as the carbon skeleton for the synthesis of glutamate. Leaf ammonium (NH_4_^+^) derives primarily from root nitrate (NO_3_^−^) assimilation and transport aboveground where it is reduced and incorporated to glutamate and glutamine. Other amino acids including aspartic acid and asparagine are synthesized by using NH_4_^+^ donated from glutamate and glutamine. These amino acids then provide the basic building blocks for protein synthesis.

**Figure 8 ijms-24-15509-f008:**
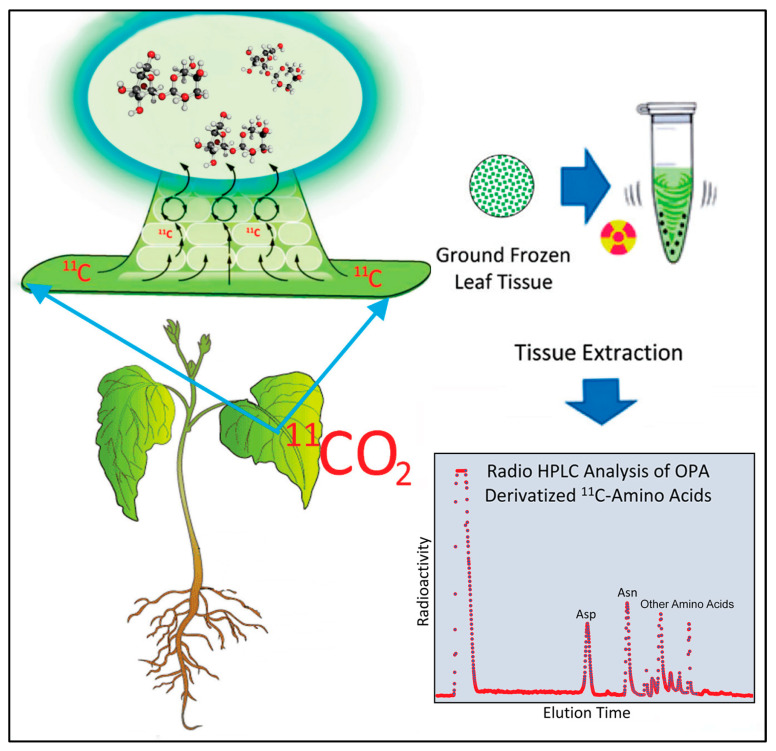
Visual workflow for [^11^C]-radiotracer analysis. Radioactivity data (red dots) for [^11^C]-amino acids was collected at a 1 Hz sampling rate.

## Data Availability

All data needed to evaluate the conclusions in the paper are present in the main text.

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
