# Peer review of "Radiocarbon Flux Measurements Reveal Mechanistic Insight into Heat-Stress Induction of Nicotine Biosynthesis in Nicotiana attenuata"

_ijms, 2023, doi:10.3390/ijms242115509_

Round 1

Reviewer 1 Report

General comments:
This manuscript focuses on an interesting topic and you explain the rationale of your research questions sufficiently. The biosynthesis of nicotine is complex and the applied carbon isotope tracer studies yielded interesting insights into the HT effects on assimilation and the biosynthesis of amino acid precursors. HT affects both negatively, more or less as expected, but nicotine levels rise. You argue that amino acids that are recycled by protein degradation can serve as the required precursors for nicotine biosynthesis.
In my understanding, you provide no experimental that support this notion. However, I assume that this is not so easy. You mention the enzyme Aparaginase that specifically catalyses the hydrolysis of asparagine from peptides. For me, the whole proteasome is enigmatic. Enzymes that catalyze peptide breakdown can easily damage themselves. Moreover, some amino acids, which potentially could be fuelled into new proteins, could have been oxidized and thus hamper the intended functionality in the newly synthetized proteins, into which  they are integrated.
I definitely do not want to force you to present a different paper but you could outline which enzymes could be quantified by qPCR in attempts to explore the genetic regulation of your studied phenomenon further.

Special comments:
Page 2: Italicize “Nicotiana“
Whole ms: Italicize “N” (nitrogen) in all compound and enzyme names
Change µmol and µmoles to µmol
Page 8: I would use the term “assimilation rate” rather instead of “photosynthesis” because some readers might associate this term with with water cleaving and  electron transport efficacy.

Author Response

General comments and author responses (in italics):

This manuscript focuses on an interesting topic and you explain the rationale of your research questions sufficiently. The biosynthesis of nicotine is complex and the applied carbon isotope tracer studies yielded interesting insights into the HT effects on assimilation and the biosynthesis of amino acid precursors. HT affects both negatively, more or less as expected, but nicotine levels rise. You argue that amino acids that are recycled by protein degradation can serve as the required precursors for nicotine biosynthesis. In my understanding, you provide no experimental that support this notion. My co-authors and I disagree with this comment.  Our radiocarbon flux measurements clearly showed that HT stress caused a decline in de novo amino acid biosynthesis from ‘new’ carbon resources yet the endogenous pool of essential amino acids (ie., Asp and Asn) were elevated.  We therefore conducted a rigorous set of experiments to measure the effect of HT stress on the total soluble protein (TSP) content targeting the leaf tissue for these assays.  Here our data showed a clear and significant decline in TSP with increased exposure times to each HT stress (38 C and 43 C). If these amino acids are not coming from N remobilization through protein degradation, where then are the plants acquiring these metabolites?

However, I assume that this is not so easy. You mention the enzyme Aparaginase that specifically catalyzes the hydrolysis of asparagine from peptides. While ASNase action may in part act on peptides, it will also catalyze the conversion of Asn to Asp in the free amino acid pool. The fact that we see a huge spike in the root Asn levels for short exposures at 38 C suggests that the temperature dependence for ASNase activity may “lag” behind that for plant protein degradation. In the edited manuscript, we’ve outlined a set of experiments for future research which will leverage ASNase overexpressing tobacco plants – here, we expect there will not be a “lag” in ASNase response converting Asn to Asp at 38 C and thus do not expect to observe an initial decline in foliar nicotine with short exposures.

For me, the whole proteasome is enigmatic. Enzymes that catalyze peptide breakdown can easily damage themselves. Moreover, some amino acids, which potentially could be fueled into new proteins, could have been oxidized and thus hamper the intended functionality in the newly synthetized proteins, into which they are integrated.

I definitely do not want to force you to present a different paper but you could outline which enzymes could be quantified by qPCR in attempts to explore the genetic regulation of your studied phenomenon further. While I agree that examining different enzyme activities in the nicotine pathway via qPCR could add value to interpreting the effects of HT stress on nicotine biosynthesis, this is not something our laboratory is equipped to do. However, see our comment above where we plan future studies using transgenic tobacco plants to further examine ASNase enzyme function.

Special comments:

Page 2: Italicize “Nicotiana“    Corrected

Whole ms: Italicize “N” (nitrogen) in all compound and enzyme names.  Corrected

Change µmol and µmoles to µmol.   Corrected both in text and in Fig. 5 axis labels

Page 8: I would use the term “assimilation rate” instead of “photosynthesis” because some readers might associate this term with water cleaving and electron transport efficacy.  Corrected

Reviewer 2 Report

The study of the effect of abiotic stress on plant metabolism is a timely research.

This research concerns the field of plant secondary metabolites in a broad sense. More precisely, the authors investigate the effect of high temperature (HT) stress on the nicotine biosynthesis via the arginine pathway. They use carbon radioisotope in pulse-shape experiments to follow the fixation of carbon in total fraction and soluble fraction of leaf proteins as well as in the total amino acid, aspartic acid and asparagine. Traditional methods are used to quantify the amount of nicotine, total proteins and Rubisco in plant tissues (leaves and roots).

The data are clearly presented. The discussion is however deceptive, mainly because there is no rational interpretation of the data shown in the manuscript and most of the conclusion are not supported by data.

Major comments.

P.2. Figure1. Add a reference in the figure caption

P.4. Figure 2. Information about the fitting is missing. The authors must provide in the figure caption 1) the equation used to fit the data and the value of the constants; 2) the goodness of the fit.

P.5 §2.2. ’… photosynthetic machinery…’ this work do not study the effect of HT stress on the photosynthetic machinery but on the carbon fixation into leaf proteins

P.6 §2.4. Figure 4. This figure shows data for asparagine. The authors must justify the interest in the study of asparagine. If arginine could influence the synthesis of nicotine then this should be mentioned in the introduction. Alternatively, the reason can be written in this paragraph.

P.6. How root were labelled with the radio isotope is not clearly described in the manuscript

P.8 Discussion

A) ‘The fact that we observed an initial decline in foliar nicotine levels at moderate 38 °C temperatures with short exposures to the stress, followed by significant increases in foliar nicotine with longer exposures tells us that the regulation of nicotine biosynthesis is complex involving several regulating factors.’

This trivial sentence indicates that the authors have no idea to interpret their data.

B) ‘Typically, sequestration of nitrogen (N) in higher plants is balanced by the plant’s C input or photosynthetic capacity [35]. This behavior has been clearly demonstrated in our past radiotracer experiments where young poplar saplings were tested for radioactive 13NO3- acquisition using Positron Emission Tomography [36]. In these studies, 13NO3- acquisition was examined along with leaf photosynthesis, measured by infrared gas exchange, and showed that changing trends in C input were tightly coupled with 13NO3- acquisition, and vice versa. Hence a lower photosynthetic capacity with HT stress should manifest in reduced N availability via nitrate uptake.’

This paragraph is out of the subject and the last sentence is purely speculative because no data on the nitrate availability and/or uptake as not been investigate at the different temperatures tested.

C) ‘Here, 38 °C HT stress did not influence the endogenous pool of leaf aspartic acid, although the 43 °C stress did elevate its content significantly relative to controls. Furthermore, the pool of leaf asparagine was significantly elevated with application of both HT stresses. Since de novo biosynthesis of these amino acids were downregulated with HT stress, there needs to be another source maintaining and even increasing their content in tissues.’

This trivial conclusion is purely speculative. Here again the authors provide no rational interpretation of their data.

Conclusion. The authors failed to explain their data based on our current knowledge of  plant physiology changes related to HT stress. Instead of providing a discussion they write a trivial setence suggesting that their must be an unidentified mechanism or regulatory process.

This difficulty to discuss their data most probably occurs because some additional experiments are required to interpret the data on a rational basis. Notably, information about the  N/C coupling is missing, the activity of the Rubisco at the different experimental condition seems to be essential.

 Clearly, analysis of arginine, asparagine and the amount of Rubisco is not sufficient to grasp underlying mechanisms of metabolic regulation in response to HT stress in Nicotiana. The authors should envision performing metabolomics to identify a large scale of metabolites and then provide an analysis of the mechanisms involved in HT stress. Other omics approaches should also be envisioned to have a broad knowledge of the different biochemical pathways that might affect the nicotine synthesis.

Author Response

General Comments and Author Responses (in italics)

The study of the effect of abiotic stress on plant metabolism is a timely research.

This research concerns the field of plant secondary metabolites in a broad sense. More precisely, the authors investigate the effect of high temperature (HT) stress on the nicotine biosynthesis via the arginine pathway. They use carbon radioisotope in pulse-shape experiments to follow the fixation of carbon in total fraction and soluble fraction of leaf proteins as well as in the total amino acid, aspartic acid and asparagine. Traditional methods are used to quantify the amount of nicotine, total proteins and Rubisco in plant tissues (leaves and roots).

The data are clearly presented. The discussion is however deceptive, mainly because there is no rational interpretation of the data shown in the manuscript and most of the conclusions are not supported by data.  I think the biggest problem the reviewer had here was the fact that we inferred (based on our prior published work Ref 36) that with inhibition of C input there would be a reduction in N uptake to maintain C/N balance. Since submitting the first draft of our manuscript for review we conducted a small set of studies using radioactive 13NO3- to provide clear evidence that reduction in C input causes a reduction in nitrate uptake.  Our work in Ref 36 demonstrates that the corollary to this is also true – that growing plants under nitrate deficiency causes a reduction in photosynthetic capacity.  With this new data in hand (see revised Fig. 3B) our conclusions in the discussion section are no longer deceptive.

Major comments.

P.2. Figure1. Add a reference in the figure caption.  Reference 21 added

P.4. Figure 2. Information about the fitting is missing. The authors must provide in the figure caption 1) the equation used to fit the data and the value of the constants; 2) the goodness of the fit. Figure 2 was revised to display in each panel the polynomial equation used to generate the trend lines with the goodness of fit presented as R2.

P.5 §2.2. ’… photosynthetic machinery…’ this work do not study the effect of HT stress on the photosynthetic machinery but on the carbon fixation into leaf proteins.  We deleted ‘photosynthetic machinery’ and replaced text with CO2 fixation which is what our experiments measured.

P.6 §2.4. Figure 4. This figure shows data for asparagine. The authors must justify the interest in the study of asparagine. If arginine could influence the synthesis of nicotine then this should be mentioned in the introduction. Alternatively, the reason can be written in this paragraph.

P.6. How root were labelled with the radio isotope is not clearly described in the manuscript

P.8 Discussion

  1. A) ‘The fact that we observed an initial decline in foliar nicotine levels at moderate 38 °C temperatures with short exposures to the stress, followed by significant increases in foliar nicotine with longer exposures tells us that the regulation of nicotine biosynthesis is complex involving several regulating factors.’

This trivial sentence indicates that the authors have no idea how to interpret their data. We deleted this sentence since the reviewer had issue with it.

  1. B) ‘Typically, sequestration of nitrogen (N) in higher plants is balanced by the plant’s C input or photosynthetic capacity [35]. This behavior has been clearly demonstrated in our past radiotracer experiments where young poplar saplings were tested for radioactive 13NO3- acquisition using Positron Emission Tomography [36]. In these studies, 13NO3- acquisition was examined along with leaf photosynthesis, measured by infrared gas exchange, and showed that changing trends in C input were tightly coupled with 13NO3- acquisition, and vice versa. Hence a lower photosynthetic capacity with HT stress should manifest in reduced N availability via nitrate uptake.’

This paragraph is out of the subject and the last sentence is purely speculative because no data on the nitrate availability and/or uptake has not been investigated at the different temperatures tested.  Note that with the newly added data on our 13NO3- experiments (see Fig 3B), we revised all the text in this part of the Discussion. See below.

Page 8 Discussion

“In agreement with past HT stress studies [31-34], our work showed that leaf CO2 assimilation decreased significantly with applied stresses. Typically, assimilation of nitrogen (N) in higher plants will be balanced by the plant’s C input or photosynthetic capacity [35]. This behavior was clearly demonstrated in the present work where HT stress reduced 13NO3- uptake in plants. As a corollary to this, our past work has also demonstrated that N-limitation during growth will downregulate leaf photosynthesis [36]. Thus, with the decreased availability of ‘new’ C and N resources, it is not so surprising that our radiocarbon flux measurements showed a significant reduction in radiolabeled amino acids.

Shown pictorially in Figure 7, the reduction in CO2 fixation seen in the present work with applied HT stress caused a rebalancing of plant resources and a reduction in NO3- uptake. Taken together, this rebalancing of resources caused the reduction in the de novo biosynthesis of essential amino acids that serve nicotine biosynthesis.”

  1. C) ‘Here, 38 °C HT stress did not influence the endogenous pool of leaf aspartic acid, although the 43 °C stress did elevate its content significantly relative to controls. Furthermore, the pool of leaf asparagine was significantly elevated with application of both HT stresses. Since de novo biosynthesis of these amino acids were downregulated with HT stress, there needs to be another source maintaining and even increasing their content in tissues.’ This trivial conclusion is purely speculative. Here again the authors provide no rational interpretation of their data. See our re-written text below.

Page 10 Discussion

“Even so, the rapid turnover of protein to free up α-amino acids at both HT stresses does not explain the initial drop observed in foliar nicotine content with short exposures at the lower 38 °C temperature. Hence, there must be another regulatory control mechanism underpinning the process.”

Conclusion. The authors failed to explain their data based on our current knowledge of plant physiology changes related to HT stress. Instead of providing a discussion they write a trivial sentence suggesting that there must be an unidentified mechanism or regulatory process. As noted before, we have added new data on plant physiological responses to HT stress relative to N uptake. Having that additional data removes a lot of the speculation this reviewer had issue with. Our work clearly shows that with the decline in C/N input with HT stress there is a decline in de novo amino acid biosynthesis (as clearly demonstrated from our radiocarbon flux measurements).  This difficulty discussing their data most probably occurs because some additional experiments are required to interpret the data on a rational basis. Notably, information about the N/C coupling is missing, the activity of the Rubisco at the different experimental condition seems to be essential. We added new data on HT stress effect on N input which should satisfy the reviewer.  Additional studies addressing Rubisco activity are not pertinent to this study – our data clearly showed that HT stress reduced 11CO2 fixation – the mechanisms underpinning this result are unimportant to our understanding nicotine biosynthesis. What is important is the fact there is a reduction in C input.

 Clearly, analysis of arginine, asparagine and the amount of Rubisco is not sufficient to grasp underlying mechanisms of metabolic regulation in response to HT stress in Nicotiana. The authors should envision performing metabolomics to identify a large scale of metabolites and then provide an analysis of the mechanisms involved in HT stress. Other omics approaches should also be envisioned to have a broad knowledge of the different biochemical pathways that might affect nicotine synthesis. These are all valid suggestions, especially applying untargeted metabolomics analyses to better understand effects of HT stress plant metabolism, but these kinds of studies are not something we can perform in the immediate future without extensive funding.

Reviewer 3 Report

Manuscript ID: ijms-2653506 - Review Report

Title: Radiocarbon Flux Measurements Reveal Mechanistic Insight into Heat Stress Induction of Nicotine Biosynthesis in Nicotiana attenuata

 Summary and opinion

The manuscript entitled ‘Radiocarbon Flux Measurements Reveal Mechanistic Insight into Heat Stress Induction of Nicotine Biosynthesis in Nicotiana attenuata' describes the nicotine biosynthesis in reaction to moderate (38°C) or severe (43°C) heat stress. Well watered plants were exposed to heat stress and different exposure times were studied.

The nicotine content in mature leaves depended on temperature and exposure time and was generally increasing during heat stress. Also carbon fixation, leaf amino acids, protein content were studied. By using radioactive CO2 pathways of carbon fixation and mobilization were studied.

 The topic of the manuscript i.e. heat stress is of importance as heat stress on plants due to global warming will increase in future and severely impact crop production. The manuscript is generally well prepared. The figures are clear and well prepared. The discussion should be more focused on the presented results (see specific comments).

 General comments:

 Figure 1 should be improved:

-          The y-axis (nicotine content) should have the same scale in all panels. Probably the scale was adjusted in a way that the nicotine content trend is easy to read. But at least for panels 1A D, and 1 B E the same scale should be used

-          Was the control data measured only at the begin of the experiment or also during the different exposure times? Something should be added in the text. Can it be excluded that there were no fluctuations over time in the control plants?

-          Figure legend of Fig.1 “average values”: it should be specified if the values are mean values

In the results it should be mentioned if there was some visible damage on leaves during heat exposure at the two temperatures or thereafter. In order to understand if the applied heat stress caused visible damage to the plants.

 Were some differences visible in plant development during heat stress exposure between the control and the two heat treatments? Like differences in plant height or leaf formation?

 Could it be possible that the reported nicotine reaction is specific for the species N. attenuata as adaptation to fire ecology, or do you think these reactions are general for nicotine biosynthesis in plants? Please add something about this in the discussion

 Sections Conclusion:

 -          the special fire ecology of N. attenuata should be mentioned here or in the Introduction/Discussion adding some citations

-          naturally heat stress occurs mostly in combination with drought stress, this stress combination is a different situation in relation to the data presented here working with well-watered plants, this possible difference should be mentioned in the discussion

Specific comments

See attached pdf file with comments.

Author Response

General Comments and Author Responses (in italics)

 Summary and opinion

The manuscript entitled ‘Radiocarbon Flux Measurements Reveal Mechanistic Insight into Heat Stress Induction of Nicotine Biosynthesis in Nicotiana attenuata' describes the nicotine biosynthesis in reaction to moderate (38°C) or severe (43°C) heat stress. Well-watered plants were exposed to heat stress and different exposure times were studied.

The nicotine content in mature leaves depended on temperature and exposure time and was generally increasing during heat stress. Also, carbon fixation, leaf amino acids, protein content were studied. By using radioactive CO2 pathways of carbon fixation and mobilization were studied.

 The topic of the manuscript i.e. heat stress is of importance as heat stress on plants due to global warming will increase in future and severely impact crop production. The manuscript is generally well prepared. The figures are clear and well prepared. The discussion should be more focused on the presented results (see specific comments).

 General comments:

 Figure 1 should be improved:  I believe the reviewer meant Fig. 2 should be improved.  We revised Fig 2 so y-axis scales are the same for panels 2 A&D and 2 B&E.

-          The y-axis (nicotine content) should have the same scale in all panels. Probably the scale was adjusted in a way that the nicotine content trend is easy to read. But at least for panels 1A D, and 1 B E the same scale should be used.

-          Was the control data measured only at the beginning of the experiment or also during the different exposure times? Something should be added in the text. Can it be excluded that there were no fluctuations over time in the control plants?  Text in the methods section was slightly revised to clarify that we used age matched plants and that no control plants were tested for nicotine a timepoint zero. 

-          Figure legend of Fig.1 “average values”: it should be specified if the values are mean values.  Corrections were made.

 In the results it should be mentioned if there was some visible damage on leaves during heat exposure at the two temperatures or thereafter. In order to understand if the applied heat stress caused visible damage to the plants. Additional text was added stating that there was no visible leaf damage from applied treatments.  See below.

 On Page 11

“Seeds of Nicotiana attenuata Torr. ex S. Watson (Accension W6 59297; USDA Agricultural Research Services, Pullman, WA, USA) were germinated in ProMix (Strawberry Hill Farms, Columbia, MO, USA) and after 10 days seedlings were transplanted to individual 3-inch pots filled with ProMix and supplemented with Osmocote™ Smart Release Plant Food (Scotts Co., Marysville, OH, USA) for continued growth in a commercial growth chamber (Model PGC-15, Percival Scientific, Perry, IA, USA) under 12-hr photoperiods at 500 μmol m-2 s-1 light intensity at constant temperature of 25 °C and relative humidity (RH, 60%). Control and heat stress studies were conducted in 5-wk-old plants. When applying heat stress, plants were transferred from their control growth chamber to a second chamber that was operated either at 38 °C or 43 °C while maintaining the same light intensity and relative humidity as the control chamber. Stress treatment duration was for 24, 48, 72 or 96-hr. Pots were placed in trays that held a surplus of water that was wicked up through the ProMix to maintain a constant well-watered status throughout the stress treatment. During treatment, plant leaves did not exhibit visible signs of damage, or necrosis.”

 Were some differences visible in plant development during heat stress exposure between the control and the two heat treatments? Like differences in plant height or leaf formation?  There really were not measurable differences in growth morphology given the short 5 day period for the full exposure cycle.

 Could it be possible that the reported nicotine reaction is specific for the species N. attenuata as adaptation to fire ecology, or do you think these reactions are general for nicotine biosynthesis in plants? Please add something about this in the discussion.  I am well aware of the rich history of research on N. attenuata regarding the effects of smoke from wildfire events on stimulating seed germination. Many bioactive chemicals in smoke have in fact been identified as stimulants for seed germination.  Hence the nickname “firechasers” was given to this species since often these plants would spring up in great numbers after a fire event. I am reluctant to add additional text on this subject since seed germination is not pertinent to alkaloid biochemistry in mature plants.

 Sections Conclusion:

 -          the special fire ecology of N. attenuata should be mentioned here or in the Introduction/Discussion adding some citations.  I and the co-authors feel this subject would detract from the focus of our paper.  However, we did expand on the ecological implications discussed in the Conclusions where initially we talked about HT stress positioning plants to better defend themselves against attack by feeding herbivores while in a weakened state of fitness.  An alternatively, HT induction of nicotine biosynthesis might also serve to promote pollination when in the reproductive stage.  Evidence in the literature suggests that pollinators (hummingbirds and bees) will respond to low levels of nicotine in floral nectar causing them to visit flowers with higher frequency. Please see the text below that was added to the revised manuscript – also note that two additional references on this subject were included.

 “From an ecological perspective, the stresses imposed by climate change will naturally weaken a plant. To survive, plants might elevate their defense posture. At least for N. attenuata, we see an upregulation in alkaloid biosynthesis with increased distribution to vegetative tissues that can help protect the plant from chewing herbivores [11-14] while in a weakened state of a C/N resource deficiency. However, an ecological paradox becomes evident considering other published works that have shown that pollinator behavior can be altered by low levels of nicotine in floral nectar. Here researchers have shown that there is an increased frequency of visitation to flowers containing small amounts of nicotine by hummingbirds [55] and bees [56] which could promote plant reproduction by boosting pollination. This begs the question whether nicotine producing plants do so exclusively to defend themselves against attack, or in this case where HT stress reduces plant fitness, serves as a mechanism to promote the survivability of their progeny?”

 -          naturally heat stress occurs mostly in combination with drought stress, this stress combination is a different situation in relation to the data presented here working with well-watered plants, this possible difference should be mentioned in the discussion.  While I agree that heat stress and drought stress naturally occur together you must note that we specifically designed our studies (ie., using well-watered plants) to avoid confounding our interpretations because of multiple stresses.  Adding a drought component, or examining the effect of drought on nicotine biosynthesis as an isolated stress is something that could, and should be considered in future studies should we receive funding to continue the research.

Specific comments - See attached pdf file with comments.  We addressed all the minor comments noted in the pdf file with the exception of adding a discussion regarding the fire ecology of N. attentuata for reasons stated above.

Round 2

Reviewer 2 Report

The manuscript as been significantly improved and is acceptable  for publication.